# Prevalence of Fox Tapeworm in Invasive Muskrats in Flanders (North Belgium)

**DOI:** 10.3390/ani12070879

**Published:** 2022-03-30

**Authors:** Emma Cartuyvels, Tim Adriaens, Kristof Baert, Frank Huysentruyt, Koen Van Den Berge

**Affiliations:** Research Institute for Nature and Forest, Wildlife Management and Invasive Species, Havenlaan 88, 1000 Brussels, Belgium; tim.adriaens@inbo.be (T.A.); kristof.baert@inbo.be (K.B.); frank.huysentruyt@inbo.be (F.H.); koen.vandenberge@inbo.be (K.V.D.B.)

**Keywords:** muskrat, fox tapeworm, invasive alien species, zoonosis, *Echinococcus multilocularis*

## Abstract

**Simple Summary:**

Fox tapeworms are pathogens that need two hosts to complete their life cycle. The reproductive part of the cycle takes place in foxes and the intermediate step usually occurs in rodents. Muskrats can also act as the intermediate host for fox tapeworms. This increases the total number of potential hosts which can also increase the number of pathogens in the environment. In Belgium, where muskrats are non-native, this is one of the ways in which an invasive alien species can have a negative effect on its new environment. From 2009 to 2017 all muskrats caught in Flanders and across the border with Wallonia and France were collected and dissected to estimate how often they were infected with fox tapeworms. Visual examination of the livers of 15,402 muskrats revealed 202 infected animals (1.31%). In Flanders, we found 82 infected animals out of 9421 (0.87%). The percentage of infected animals did not increase during the research period. All the infected animals in Flanders were found in municipalities along the Walloon border. We did not observe a northward spread of EM infection from Wallonia to Flanders.

**Abstract:**

One way in which invasive alien species affect their environment is by acting as pathogen hosts. Pathogens limited by the availability of the native host species can profit from the presence of additional hosts. The muskrat (*Ondatra zibethicus*) is known to act as an intermediate host for the fox tapeworm (*Echinococcus multilocularis*). From 2009 to 2017, 15,402 muskrats caught in Flanders and across the border with Wallonia and France were collected and dissected with the aim of understanding the prevalence of this parasite in muskrats. Visual examination of the livers revealed 202 infected animals (1.31%). Out of the 9421 animals caught in Flanders, we found 82 individuals (0.87%) infected with *E. multilocularis*. No increase in prevalence was observed during this study. All of the infected animals in Flanders were found in municipalities along the Walloon border. We did not observe a northward spread of *E. multilocularis* infection from Wallonia to Flanders. We hypothesise that the low prevalence is the result of the reduced availability of intermediate hosts and the successful control programme which is keeping muskrat densities in the centre of the region at low levels and is preventing influx from other areas. Our results illustrate that muskrats are good sentinels for *E. multilocularis* and regular screening can gain valuable insight into the spread of this zoonosis.

## 1. Introduction

Invasive Alien Species (IAS) represent a significant source of human-mediated introduction of pathogens to a new host or region [1]. They can facilitate the spread of infectious diseases, either by acting as an alternative host for native pathogens or by bringing in pathogens from their native range. These pathogens can qualify as emerging infectious diseases (EIDs), some of which form a significant threat to public health [2]. When an invasive species becomes an alternative host for native parasites this leads to either spill back, where the presence of the additional host increases disease impacts in native species [3], or dilution, where more diverse host communities inhibit the spread of parasites [4]. Well known examples of diseases disseminated by IAS or exotic pet trade are chytridiomycosis in amphibians, squirrel pox, and crayfish plague [5,6,7].

The muskrat (*Ondatra zibethicus*) is a rodent native to swamps and wetlands in North America and an invasive alien species (IAS) in Europe. The species was first introduced in 1905 in Europe as a furbearer [8] and has spread to suitable habitats in most European countries where they cause damage to dykes and agricultural crops. Muskrats can also be carriers of diseases such as tularemia and leptospirosis and they can carry parasites, such as tapeworms, most commonly *Taenia taeniaeformis* [9]. Because of their economic, human health, and environmental impacts, muskrats are managed in several European countries but most intensively in Belgium and the Netherlands. Management practice consists mostly of systematic trapping to reduce muskrat densities, lowering the levels of muskrat damage [10]. In Flanders, an intensification of the trapping effort since 2000 resulted in a spectacular decline of the muskrat population, as shown by the number of muskrats caught per year: from over 60,000 animals in 2001 to just over 5000 in 2020 [11].

Fox tapeworm (*Echinococcus multilocularis*) (EM) is a parasite of which the adult stage lives in the intestines of foxes and occasionally other carnivores [12]. The adult tapeworm is only a few millimeters in size and consists, on average, of five proglottids or segments [13]. Fox tapeworms incubate for 26 to 29 days, after which they produce up to 100,000 eggs per day [14]. This can last up to four months [15]. Once the eggs are released into the environment, they can be ingested by intermediate hosts. Several mammal species can act as intermediate hosts for EM, including the European beaver (*Castor fiber*), brown hare (*Lepus europaeus*) and coypu (*Myocastor coypus*) [16]. However, voles (*Arvicola* sp., *Microtus* sp.) and muskrats are considered the main intermediate hosts in Europe [15,17].

Muskrats ingest the infective eggs through contaminated food and vegetation. The degree to which the species is infected indicates the presence of eggs in the environment and thus the potential risk to humans. It can be considered as a bioindicator for the presence of EM in new areas [18,19]. Moreover, muskrats show a greater chance of infection than the naturally present intermediate hosts and can therefore function very well as sentinel species [20].

*Echinococcus multilocularis* can develop into a life-threatening infection in humans, alveolar echinococcosis (AE), and is a species of great concern in the context of public health. Humans can act as an aberrant host once they become infected with the eggs of the parasite through contact with infected animals or their faeces. One of the infection routes is contaminated food such as mushrooms, wild berries, or vegetables from gardens. Most infections occur in people active in forestry, horticulture, agriculture, or hunting, but also pet owners [15,21,22]. The eggs will develop to alveolar hydatid cysts, mainly in the liver, leading to AE. Due to the strong increase in red fox (*Vulpes vulpes*) populations in Europe at the end of the last century [23,24,25], the concern for this serious parasitosis also grew [13,15]. In 2020 there were 115 confirmed cases of AE in the EU, 10 of those were from Belgium [26].

Southern Germany, Switzerland, and Alpine Austria are historically considered the endemic area for this infection in Europe [27]. This area has been expanding to the north, west, and east since the 1970s [28]. For example, Combes et al. [29] observed a strong westward spread of the fox tapeworm in red foxes in France. In the Dutch province Limburg, a northward expansion was found moving at a speed of 2.7 km per year [30]. Vervaeke et al. [31] also found a northwestward spread of infected foxes in Belgium, starting from the south of Belgium. However, they also observed that in 2000–2002 this spread stabilized at the regional border. Other studies have also confirmed this difference in the prevalence of EM in foxes between Flanders (northern Belgium) and Wallonia (southern Belgium): Wallonia has a prevalence between 5 and 25%, while the prevalence in Flanders ranges between 0 and 1% [32]. Most infected red foxes found in Flanders are found close to the regional southern border [33,34].

From 1994 onward, muskrats from Flanders have been autopsied with the aim of understanding their demography. In 2008, we first detected EM in a muskrat from Lessines (Wallonia, Belgium), just across the border with Flanders. Therefore, from 2009 onward, all available muskrats were collected and examined with the aim of answering the following questions:What is the prevalence of EM in muskrats captured in Flanders?Does this prevalence show an increase over time?Does the geographical range of EM infected muskrats expand?

## 2. Materials and Methods

Muskrats were systematically collected by the Flanders Environment Agency, the main trapping organization in Flanders, in the period from 2009 to 2017. Additionally, muskrats caught just across the border in Wallonia and France were also examined, since this was an area of particular interest. During this period, 17,497 muskrats were caught by the Flanders Environment Agency (Figure 1). In the same period, 15,402 muskrats were dissected and scored for EM. Thus, 12% of all specimens caught during this period were missing. These were animals that had decayed too much or animals to which no trapping location could be assigned.

All reported infection rates are based on dissected animals caught in Flanders between 2009 and 2017 unless stated otherwise. More general results, i.e., monthly weight distribution, are based on the total dataset collected between 1994 and 2018.

Animals were stored at −20 °C pending autopsy to neutralize the protoscoleces and/or germinative cells of the EM metacestode. Animals were then cleaned of dirt (mud, leaves, stones, etc.) and excess water in the fur. Each animal was weighed, sexed, and examined for injuries or particularities. We checked the abdominal cavity for the presence of macroparasites. This involved screening the liver surface for irregularities, discolourations, and lesions. We scored the EM presence as positive, negative, or inconclusive. In most cases the presence of EM was clearly recognizable, however, when this was not the case, samples of the presumed infected liver tissue were taken for confirmation and examined under a light microscope to check for the presence of protoscoleces.

Data were grouped by year, sex, month, and weight, and the significance of observed differences between groups was tested using a chi-square test or a Fisher’s Exact Test if subgroups were too small. The data were analysed using R [35].

## 3. Results

Visual inspection of 9421 livers revealed 82 infected animals (0.87%, CI 0.70–1.01%). In all regions combined, we found 202 infected animals out of 15,402 (1.31%, CI 1.14–1.50%), with no infections occurring in the 2477 animals trapped in France. The parasite mainly occurred in the western part of the Flanders-Wallonia border (Figure 2) and was found in muskrats from this area yearly. As of 2013, positive cases were also found in the eastern part of the border, with the most eastern observation occurring in 2017 in Tongeren. All cases were found along the administrative border with Wallonia, no northward spread was observed.

The overall prevalence of EM in muskrats fluctuated around an average of 0.9% (range 0.2–1.9%). Visual interpretation of these results showed no up- or downward trend over the years (Figure 3) and there was no significant correlation between year and prevalence (cor = −0.34, CI = −0.82–0.41, *p* = 0.36).

### 3.1. Sex Ratio and Infection Rate

We find a slightly higher number of infected animals among males compared to females (Figure 4), however there was no significant effect of sex on infection rate (Χ^2^ = 1.21, df = 1, *p*-value = 0.27).

### 3.2. Weight/Age Distribution and Infection Rate

The rate of infection in muskrats under 500 g was particularly low in comparison with heavier/more adult animals (Figure 5). The number of inconclusive cases also increases with weight (Figure 5). There was a significant effect of weight on infection rate (*p*-value < 0.005).

### 3.3. Infection Rate and Month

The highest percentage of infected muskrats was found in spring (Figure 6). We found a significant effect of month on infection rate (Χ^2^ = 144.89, df = 11, *p* < 0.005). Given the seasonal reproduction of the muskrat, the age structure of the population changes during the year. The percentage of young animals increases drastically from May on, while fewer young are born in autumn. The spring population consists mainly of subadults and a smaller fraction of older animals (>1 year) (Figure 7). This could explain the seasonal difference in prevalence.

## 4. Discussion

The prevalence of EM in Flemish muskrats fluctuated around 0.9%, indicating low infection rates in the region. The prevalence also remained fairly constant throughout the sampling period (2009–2017). *Echinococcus multilocularis* positive muskrats were mainly caught in the western part of Flanders, yet additional positive cases emerged in the eastern province of Limburg. Overall, in Flanders, the parasite was found only in muskrats caught close to the regional border.

Using body weight as a proxy for age (based on our findings in Figure 7), older muskrats were found to have higher infection rates than younger animals. From weaning, a young muskrat runs the risk of becoming infected with tapeworm eggs. The probability of infection increases with the time an animal is present in an infected area. After ingestion of EM eggs, the likelihood of detecting cysts in the liver also gradually increases. Based on these two arguments, an increase in the risk of a visible infection can therefore be assumed with increasing age. We also found the number of inconclusive cases increasing with weight. This may be due to an accumulation of other parasites in the liver. For example, it appears that there is also a positive correlation between the body weight of the muskrat and the degree of infection with *T. taeniaeformis* (correlation = 0.49, *p* < 0.001) (unpublished results). The fact that no infected animals with a weight >1500 g were found could be explained by the fact that this is a very small cohort (75 animals), that it might be harder to distinguish EM infection from other sometimes massive infections with *T. taeniaformis*, or that infected muskrats have a higher mortality risk.

The univariable approach followed here to describe patterns of infection across years, season, and demographic parameters (sex, age) of muskrats inherently represent limitations. Although based on an extensive dataset of autopsied muskrats, in reality, the epidemiology of echinococcosis is complex, and demographic, spatial, and temporal variables might exhibit multiple interaction effects contributing to observed infection status [36]. Additionally, the effect of muskrat management has not been explicitly tested. Risk assessment for this zoonosis also ideally needs to consider all intermediate and potential end hosts, including foxes and humans. Further multivariate analysis is needed to reveal such complex interaction patterns.

Whether the parasite is permanently present in Flanders or whether most (infected) animals are migratory muskrats that have contracted the infection in Wallonia remains open for discussion. The second option seems more likely as infection rates for both foxes and muskrats are much higher in Wallonia, especially south of the Meuse River [12,32]. Muskrats also have a higher chance of reproducing and living long enough in the environment to ingest and develop the infection due to lower trapping efforts in Wallonia [37]. Muskrats migrate mainly in the spring as young adults when they look for a mate, with males migrating further than females [38]. More males are indeed caught in Flanders. We do find a similar sex ratio for additional observations from Wallonia, however, these observations were all made very close to the Flanders-Walloon border and could therefore also experience a high influx of migratory animals. When looking at muskrats collected more to the south of Wallonia [39] we see a different, more equal sex ratio (e.g., 53.1% male, 46.9% female). On the other hand, little is known about the symptoms and mortality of EM infection in muskrats. Therefore, it is unclear how likely it is that an infected animal will migrate over long distances. Examination of the natural intermediate hosts along both sites of the Flemish Walloon border could reveal this.

Vervaeke et al. [31] predicted that the expansion of fox tapeworm seen in Wallonia would continue northwards in Flanders, but this was not reported in more recent studies [33,34]. Genetic research by Knapp et al. [40] showed that the observed changes in distribution seen all over Europe were mostly the result of increased attention for the disease rather than an actual geographic expansion. This leaves the question of why the infection has never entered Flanders, especially with the reoccurrence of the red fox in recent decades. A possible explanation is that there are too few suitable intermediate hosts in Flanders to complete the cycle. Although few concrete figures are available on this, the hypothesis that *Microtus* spp. are relatively rare in Flanders [34] seems justified. According to the Flemish Red List of mammals [41], similar to the two bigger vole species, three of the four smaller vole species (Short-tailed field vole (*Microtus agrestis*), European pine vole (*Microtus subterraneus*), and common vole (*Microtus arvalis*)) are classified in the ‘near threatened’ category. In Wallonia, on the other hand, almost all small vole species are still common and there are no indications of a decline [42]. It is striking in this regard that these species are not the staple food of the fox in Flanders, as is generally the case [43]. Instead, the brown rat (*Rattus norvegicus*) is the primary prey species [34,44]. We hypothesize that successfully keeping the muskrat population down has also kept the potential pool of intermediate hosts small, thus helping to prevent the spread of the fox tapeworm.

Due to targeted, intensive control, muskrats no longer occur all over Flanders. The vast majority of the animals are caught at the regional border. Since the fox tapeworm could still be spread across Flanders in foxes and other intermediate hosts, monitoring the fox tapeworm in muskrats alone is therefore not the best proxy to gain insight into the general occurrence of the tapeworm. However, since muskrats are considered as a very good sentinel species for EM, regular screening of the muskrats in Flanders and especially near the Flemish Walloon border could still be valuable to gain insight into a possible spread of this zoonosis.

## 5. Conclusions

Flanders currently has a low prevalence of EM in muskrats and there has been no increase in prevalence during this study. The infected animals were all located in the regional border zone. Our results illustrate that muskrats are good sentinels for *E. multilocularis* and regular screening can gain valuable insight into the spread of this zoonosis.

## Figures and Tables

**Figure 1 animals-12-00879-f001:**
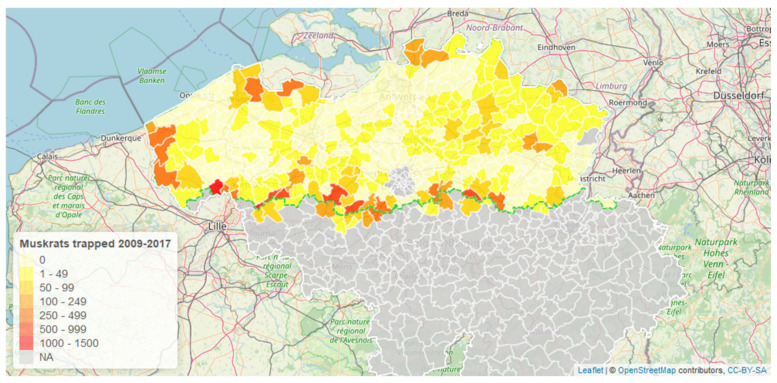
Number of muskrats trapped by the Flanders Environment Agency between 2009 and 2017. The regional border between Flanders and Wallonia is indicated by a green dashed line.

**Figure 2 animals-12-00879-f002:**
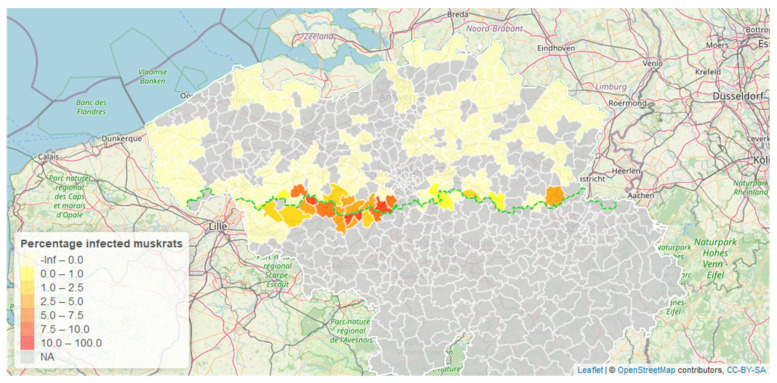
Percentage of infected muskrats out of all animals dissected between 2009 and 2017. Additional observations from Wallonia (regional border indicated by green dashed line) are also included.

**Figure 3 animals-12-00879-f003:**
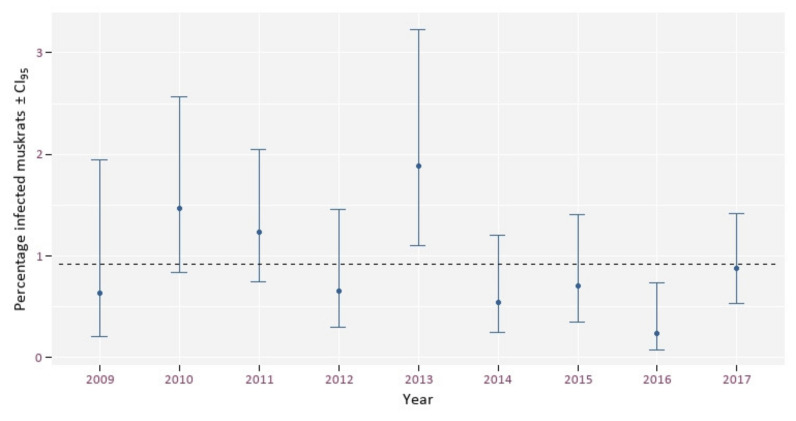
The percentage of EM infected muskrats in Flanders per year. The dashed line indicates an average infection rate of 0.9%.

**Figure 4 animals-12-00879-f004:**
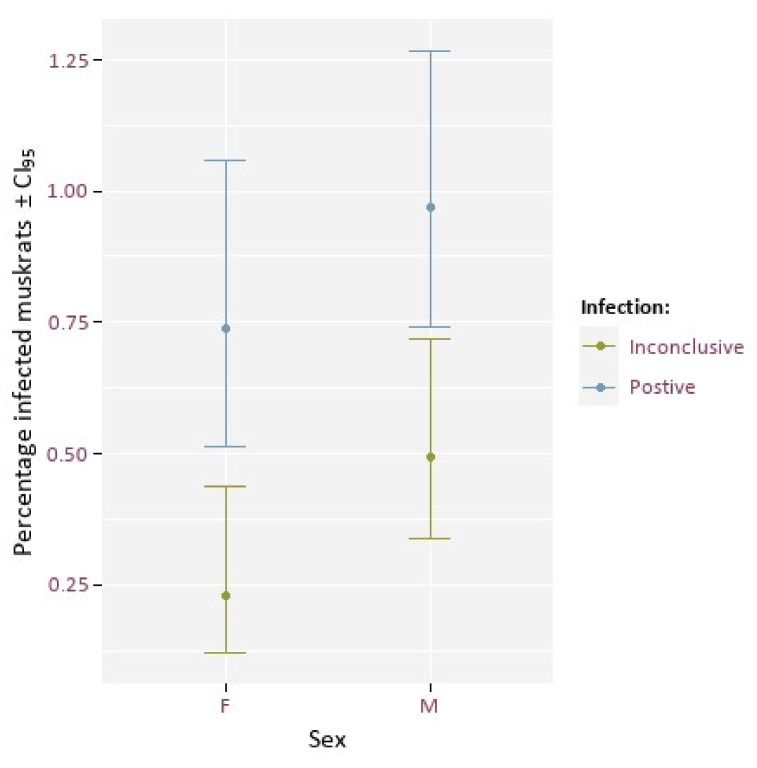
Percentage of muskrats with positive (blue) and inconclusive (green) EM presence by sex.

**Figure 5 animals-12-00879-f005:**
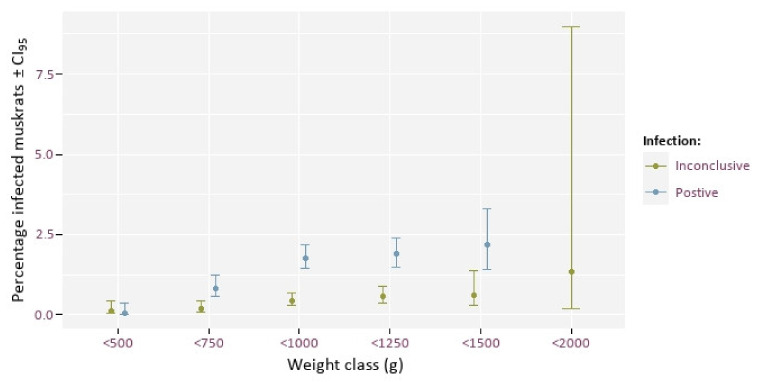
Percentage of muskrats with positive (blue) and inconclusive (green) EM presence by weight class (g).

**Figure 6 animals-12-00879-f006:**
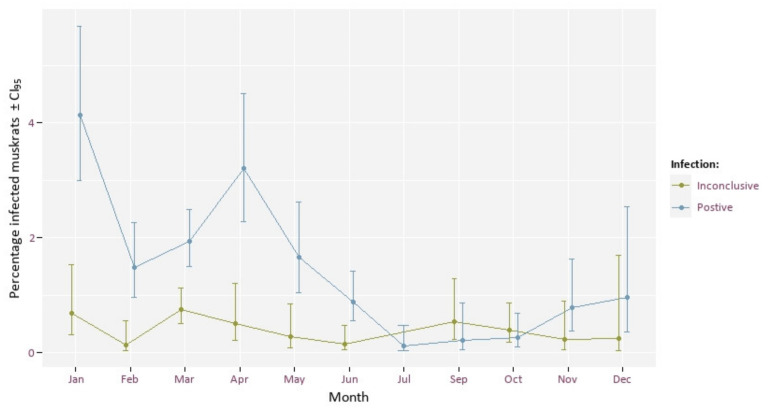
Percentage of muskrats with positive (blue) and inconclusive (green) EM presence by month.

**Figure 7 animals-12-00879-f007:**
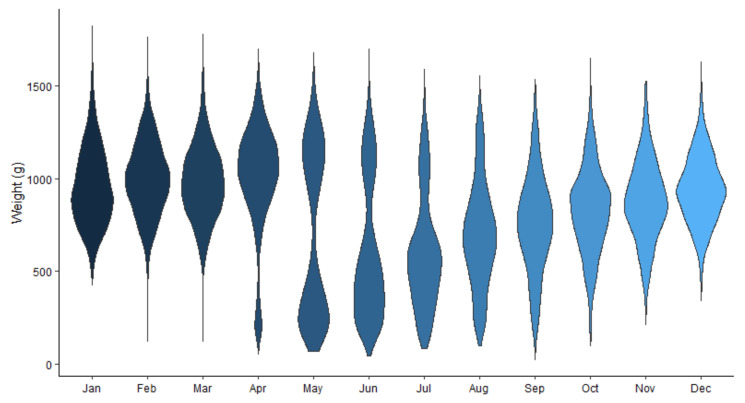
Distribution of the body weight of the dissected muskrats per month.

## Data Availability

Data analysed as part of this study can be accessed on Zenodo doi:10.5281/zenodo.6355420.

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
