# Peer review of "Prevalence of Fox Tapeworm in Invasive Muskrats in Flanders (North Belgium)"

_animals, 2022, doi:10.3390/ani12070879_

Round 1

Reviewer 1 Report

Dear Authors,

This is a nice piece of work. You have an excellent and very precious database in your hands. However, you have to analyse the data properly. A complex statistical approach should be applied to get as much as possible from the data. At this moment you report just values such as number of infected animals, weight, sex etc. 

With such a database you can easily  analyse both intrinsic and extrinsic factors that influence prevalence of EM in muskrats in a long-term approach.

Please consider contacting someone who is specialising in such analyses. After this your paper will bring novelty in understanding the ecology of EM. 

I truly encourage you to do so. 

Below I enclosed major issues: 

Sample summary:

P1 L10 „Usually occurs in voles” - please change voles into rodents

P1 L10 “Muskrats can also act as the intermediate host” - please add “..for fox tapeworms”.

Abstract:

P1 L21 “additional reservoirs” please rephrase. 

P1 L26 please rephrase the sentence to read: We investigated 9421 animals and found 82 individuals (0.87%) infected with E. multilocularis.

P1 L28 EM - please explain 

P1 L31 this is not the conclusion of your work. Please rephrase. 

Keywords: please add Echinococcus multilocularis 

Introduction:

P1L 39 Do all newly arrived pathogens are classified as EIDs?

P2 L45 Please add references for given examples

P2 L48 please consider changing EU to European

P1 L57 any possible references on these numbers?

P1 L61 please rephrase the sentence to be in an active voice. Please avoid passive voice.

P1 L65 Please consider changing Microtus arvalis into Microtus spp. Please see and consider citing Miller et al. https://doi.org/10.1016/j.ijppaw.2016.03.001

P1 L75 please consider mentioning infection rout with contaminated food like wild blueberries or mushrooms.

P1 L76 one sentence about number of human cases in Europe in 2020 or 2021 would be appreciated to show the scale of the problem

P1 L77 please consider citing Karamon et al DOI: 10.1007/s00436-013-3657-z

P1L89 Where Wallonia has a prevalence - please rephrase

P1 L94 any refs for the 1st description?

P1 L95 - L100: As I understand these questions were aims of the Environmental Agency. But, what are the aims of your work? One sentence with clear aims of your work would nicely end the Introduction. 

Material and Methods

This section is clear. However, statistical approach was not described. 

Results

Please always report the prevalence (percentage of infected animals) with +/- 95 Confidence limits. Please see Zaloumis et al 2015 -> doi: 10.1017/S0031182015000748 and Morrison 2002 10.1016/s0020-7519(02)00064-4.

P4 L138 “The percentage of infected muskrats fluctuated around an average of 0.9%” - please rephrase. Do you mean overall prevalence? 

Figure 3 - The graph should have error bars with +/-95CL for each year. Please see Grzybek et al 2019 (Fig 1) https://doi.org/10.1016/j.ijppaw.2019.03.005 for the example (you don’t have to cite this work in your ms. This is just an example). 

Have you tested the effect of year on the prevalence of EM in your population? Looking at Fig 3 there might be some differences between years. But you have to run a statistical test to asses it.

Table1 - this table shows sex ratio on muskrats - it does not present any information related to EM. This information can me moved to M&M section -> “We analyzed n number of animals (x - males and x - females)”.

Table 2 and prevalence of EM in males and females. I am not very happy with comparison of sex ratio vs presence / absence of infection. You should test if there was a difference in the number of infected animals vs non infected in both sexes. You could also check the impact of intrinsic (year of study) and extrinsic (sex of the host) factors on EM prevalence. Please see the example -> Grzybek et al 2018 (https://dx.doi.org/10.1038%2Fs41426-018-0149-3) - you do not have to cite this work in your ms. This is just an example. 

Figure 4 - Percentages should be presented with error bars +/- 95CL. 

Again you have a nice intrinsic factor here - weight of an animal. You could run a very nice test to check if the weight of animals impacted the EM prevalence. 

If you state that there is a difference between light and heavy individuals please provide statistics. 

Figure 6 - again the same. You have a very strong data set! You could present tour de force analysis of seasonality. 

Discussion

P7 L186 You have never mentioned that you are using animals' weight to indicate their age. Please describe it in M&M section. Moreover if you decide to divide all individuals into age groups please provide the weight range for the groups. 

P7 L189 how do you know it? 

P7 L192 This might be true but your argument is wrong. The heavier individual the older it is. In the case of pathogens that cause chronic infections, the likelihood of being infected and the abundance of parasite burdens increase with the age of the host. 

P7 L198  you can’t say it without a statistical test. 

P7 L224-234 - i got lost with this part. It is unclear. I don’t understand the real link between muskrats and small rodents. Please try to rephrase this part. 

Reviewer 2 Report

Authors aims to determine EM prevalence muskrats captured in Flanders, evaluating changes over time and the geographical expansion. 

Specific suggestion/comments are highlighted in the attached .pdf file. 

The topic is of interest and results helps to understand the evolution of EM infection in this intermediate host. Under this, some queries emerge, why not perform a logistic regression using weight and sex? Why not to use an analytic approach to the time series evaluation (e.g. time series additive decomposition and Lunj-Box test? that way it won't be just a visual evaluation and it would be an objective result.

In general terms the manuscript is clear and well written, easy to follow-up.

Round 2

Reviewer 1 Report

I find the ms ready for publication. 

Author Response

We thank the reviewer for their kind help in improving the manuscript.

Reviewer 2 Report

Authors aims to determine EM prevalence muskrats captured in Flanders, evaluating changes over time and the geographical expansion. 

Minor specific suggestion/comments are highlighted in the attached .pdf file. 

the manuscript has been sufficiently improved and have included almost all suggestion from reviewers.

One topic that keeps me concerned is the data analysis or statistical analysis, using chi-square or Fisher exact test does not improve the confidence or the power of the results, using a univariable approach lacks a real look at events in nature, that is why I proposed carrying out a multivariable test, and a deeper analysis regarding the temporal behavior recorded in the study.

Author Response

We thank the reviewer for their kind help in improving the manuscript. We agree that more information can be gained from performing a multivariate analysis but we do believe that descriptive and univariate analyses have merit on their own, especially in such a large database as this. 

  • 33: Altered as suggested.
  • 71: Altered as suggested.
  • 84: Altered as suggested.
  • 133: We agree with the reviewer that there is information to be gained from doing a multivariate analysis. We therefore added the following paragraph to the discussion (L228): "The univariable approach followed here to describe patterns of infection across years, season and demographic parameters (sex, age) of muskrats inherently represents limitations. Although based on an extensive dataset of autopsied muskrats, in reality, the epidemiology of echinococcosis is complex and demographic, spatial and temporal variables might exhibit multiple interactions effects contributing to observed infection status [36]. Also, the effect of muskrat management has not been explicitly tested. Risk assessment for this zoonosis also ideally needs to consider all intermediate and potential end hosts including foxes and human activity. Further multivariate analysis is needed to reveal such complex interaction patterns."